# Synergistic effect of two human-like monoclonal antibodies confers protection against orthopoxvirus infection

Hadas Tamir[1,2], Tal Noy-Porat [1,2], Sharon Melamed[1], Lilach Cherry-Mimran[1], Moria Barlev-Gross[1], Ron Alcalay[1], Yfat Yahalom-Ronen[1], Hagit Achdout [1], Boaz Politi[1], Noam Erez [1], Shay Weiss[1], Ronit Rosenfeld [1], Eyal Epstein[1], Ohad Mazor [1], Efi Makdasi[1], Nir Paran [1] & Tomer Israely [1] ✉

The eradication of smallpox was officially declared by the WHO in 1980, leading to discontinuation of the vaccination campaign against the virus. Consequently, immunity against smallpox and related orthopoxviruses like Monkeypox virus gradually declines, highlighting the need for efficient countermeasures not only for the prevention, but also for the treatment of already exposed individuals. We have recently developed human-like mono-clonal antibodies (mAbs) from vaccinia virus-immunized non-human primates. Two mAbs, MV33 and EV42, targeting the two infectious forms of the virus, were selected for in vivo evaluation, based on their in vitro neutralization potency. A single dose of either MV33 or EV42 administered three days post-infection (dpi) to BALB/c female mice provides full protection against lethal ectromelia virus challenge. Importantly, a combination of both mAbs confers full protection even when provided five dpi. Whole-body bioimaging and viral load analysis reveal that combination of the two mAbs allows for faster and more efficient clearance of the virus from target organs compared to either MV33 or EV42 separately. The combined mAbs treatment further confers post-exposure protection against the currently circulating Monkeypox virus in Cast/EiJ female mice, highlighting their therapeutic potential against other orthopoxviruses.

Smallpox, caused by the variola virus (VARV), has been a devastating and deadly human disease for centuries. However, following a global mass vaccination campaign utilizing vaccinia virus (VACV)-based vaccines, the World Health Organization (WHO) declared in 1980 that smallpox has been eradicated[1]. Despite this unprecedented achievement, there is still a risk of smallpox re-emergence, due to accidental leakage or deliberate release of VARV from either viral stocks or syn-thetically generated virus. Additionally, the emergence of zoonotic strains, such as monkeypox virus (MPXV) and cowpox virus (CPXV) poses new challenges, as evidenced by recent outbreaks[2] underscoring the need for effective antiviral drugs to combat potential outbreaks.

Smallpox vaccine provides protection when given prior to expo-sure and even post-exposure, yet, vaccines might not be valuable to immunocompromised individuals. Despite their approved efficacy, the conventional vaccines are associated with rare yet severe adverse effects in at-risk individuals. Vaccinia virus immunoglobulin (VIGIV) is a purified immune globulin preparation derived from plasma of vacci-nated donors approved for the treatment of certain post-vaccinal adverse effects. Despite the benefits of VIGIV, and due to its limited supply, price, inconsistency, and limited efficacy, efforts have been underway for over a decade to develop advanced technologies to replace it. In 2011, Lantto et al. presented the preparation of a cocktail

[1]Israel Institute for Biological Research, Ness Ziona, Israel. [2]These authors contributed equally: Hadas Tamir, Tal Noy-Porat. ✉e-mail: tomeri@iibr.gov.il

containing 26 recombinant monoclonal antibodies (mAbs) as an alternative to VIGIV[3]. This mAbs mixture protected mice when given up to 14 days before or six days post challenge[4]. Subsequently, Gilchuk et al. in 2016[5] has demonstrated the effectiveness of a cocktail containing six mAbs capable of cross-neutralizing VACV, CPXV, MPXV, and VARV, providing protection up to three days following VACV infection in mice model. This cocktail was further refined to include only four mAbs, directed against both viral forms (Mature virion; MV and Enveloped virion; EV), proving its efficacy in mice as a prophylactic measure when administered one day before vaccinia virus challenge. Additional studies have evaluated the protective effect of several other neutralizing antibodies against orthopoxviruses[6–8].

Ectromelia virus (ECTV), a member of the Poxviridae family and the causative agent of mousepox, is a natural mouse pathogen that provides a valuable small animal model for human smallpox. ECTV, similarly to VARV, encodes multiple host-specific immunoregulatory genes and causes severe lethal disease associated with multi-organ high viral loads and various manifestations[9–12]. This model is therefore considered of high value for testing vaccines and antivirals in a small animal model.

We have recently demonstrated the isolation of high affinity mAbs against orthopoxviruses which could neutralize the two infectious forms of the virus, MV and EV, in vitro[13]. In the present study, we have selected two mAbs targeting MV surface protein D8 and EV surface protein A33 and demonstrate their in vivo post-exposure potency using ECTV infected mice. A full protection was demonstrated when either of these antibodies were administered at a single dose at three dpi. Further attempts to extend their therapeutic window revealed that a combined therapy of the two mAbs, directed against both forms of the virus, conferred full protection even five dpi, with an effective and rapid viral clearance. These findings highlight a synergistic mode of action of only two human-like mAbs, targeting both viral forms, protecting mice against a virulent Poxvirus infection as late as five dpi.

## Results

### Treatment with anti D8 and anti A33 mAbs protects infected mice against a lethal dose of ECTV

We have recently isolated recombinant mAbs from a phage display library derived from rhesus macaques immunized with VACV, which were highly potent in neutralizing VACV in vitro[13]. Based on their promising neutralization potency, we selected two mAbs, MV33 (targeting MV surface protein D8) and EV42 (targeting EV surface protein A33), for further in vivo evaluation. To assess their therapeutic efficacy, BALB/c mice were infected by intranasal (i.n.) instillation with a lethal dose (50 PFU) of ECTV and treated with a single intraperitoneal (i.p.) injection of 200 μg of either MV33, EV42, or 4 mg of commercial VIGIV (Omrix) one dpi. Infected untreated mice served as a control. Morbidity (based on weight loss) and mortality were monitored for 21 days. Figure 1A shows that untreated mice developed morbidity starting 8-9 dpi, as evident by a weight loss exceeding 20% of their initial weight, leading to 85% mortality by day 12. Treatment with MV33 or VIGIV provided partial protection with survival rates of 65% and 50%, respectively. Remarkably, EV42 treatment resulted in 100% survival. Considering that both MV and EV forms contribute to viral infection in vitro[14] and in-vivo[15,16], we sought to investigate whether combining both MV33 and EV42 would improve the morbidity and survival rates of ECTV infected mice. Mice were infected and treated with a half dose (100 + 100 μg) or full dose (200 + 200 μg) of MV33 and EV42. The results depicted in Fig. 1A, F demonstrate that these combinations resulted in reduced body weight loss, faster recovery, and full protection. Based on these promising results, we extended the therapeutic window of MV33 and EV42 when provided separately or in combination (half dose each) at 2–5 dpi. In this set of experiments, an irrelevant antibody (BLN12; directed to SARS-CoV-2) was used as an isotype control (IgG). As depicted in Fig. 1B–J, BLN12 treated mice

experienced morbidity around eight dpi, resulting in a survival rate of 0–30% a few days later. VIGIV treatment conferred 85% survival and significantly ameliorated the disease when provided two dpi, but had no significant therapeutic value when provided on either day three, four or five. Notably, mice treated with 200 μg of either MV33 or, EV42, or a combination of both at two or three dpi, although exhibiting body weight loss (Fig. 1B, C), were fully protected from death (Fig. 1G, H). When extending the therapeutic window to four dpi, MV33 or EV42 treatment alone provided partial, though significant protection (80% and 65% survival, respectively), while the combination of MV33 and EV42 still conferred 100% survival (Fig. 1I). Similar trends were observed at day five pi, where treatment with either MV33 or EV42 alone resulted in body weight loss and partial protection (65% each), whereas the combination of both conferred again full protection (Fig. 1J). These results suggest a potential synergistic effect of these two antibodies by simultaneously targeting both viral infectious forms (MV and EV). When the mAb treatment using both MV33 + EV42, was provided on six dpi, no protection in terms of morbidity or survival was observed (Supplementary Fig. 1).

### Treatment with anti D8 and anti A33 antibodies inhibits ECTV replication and dissemination

To further investigate the impact of these two mAbs on virus spread throughout the disease progression, we conducted whole-body bioluminescent imaging (BLI) using a recombinant ECTV expressing firefly Luciferase (ECTV-Luc). Compared to the parental ECTV used above, ECTV-Luc is slightly less virulent[17] as seen in supplementary Fig. 2. Hence, to achieve 20 LD50, a dose of 140 PFU was used, instead of the 50 PFU used with the parental virus. Mice were infected and treated with a single dose of 200 μg of either MV33, EV42, a combination of the two (100 + 100 μg), VIGIV (VIG; 4 mg) or with an isotype control (BLN12; 200 μg) at three- or five- dpi (Figs. 2 and 3 respectively). Whole body images were performed daily, starting from 24 h after treatment (four or six dpi) until 14 dpi. The total photon flux values were quantified, and the signal/noise intensity is presented in Figs. 2B and 3B.

At four dpi, the luciferase signal in the isotype control antibody (BLN12) treated mice (Fig. 2) was evident mainly in the nasal cavity and the spleen. With time, the signal intensity increased mainly at the thoracic and abdominal regions encompassing the spleen, liver, and lungs (day eight post infection). In contrast, all MV33, EV42 or MV33 + EV42 mAbs treated mice showed a reduction in bioluminescence signal (Fig. 2B). VIGIV treatment resulted in reduced luciferase signal, yet, three out of four mice exhibited delayed clearance (day 12; Fig. 2A, B), a phenomenon not observed with our human-like mAbs. It is worth mentioning that the apparent enhancement of signal in VIGIV treated group on day 12 is due to a lower presented dynamic range ($5*10^6$–$5*10^7$ photon/s/cm$^2$/sr). As mAbs treatment ameliorated the disease and prevented mortality even when provided at later time points post infection, we also chose to examine the effect of these antibodies on viral dissemination in mice treated at five dpi. As seen in Fig. 3, mice treated with a single dose of either MV33 or EV42 exhibited a partial signal reduction as compared with the isotype control, which corresponded with their partial mortality rate (see Figs. 3 and 1J). However, the combined therapy effect was evident, and luciferase signal was reduced with time.

### Treatment with anti D8 and anti A33 antibodies reduces viral loads in target organs

To obtain a more focused and quantitative evaluation of the impact of these mAbs on viral dissemination and based on the BLI data of day eight post infection, we assessed viral loads in target organs, namely spleen, liver, and lung. Mice were infected with 50 PFU of ECTV and treated at either three or five dpi with MV33 (200 μg), EV42 (200 μg), MV33 + EV42 (100 + 100 μg), VIGIV (4 mg), or with an isotype control (BLN12; 200 μg). At eight dpi, mice were sacrificed, and

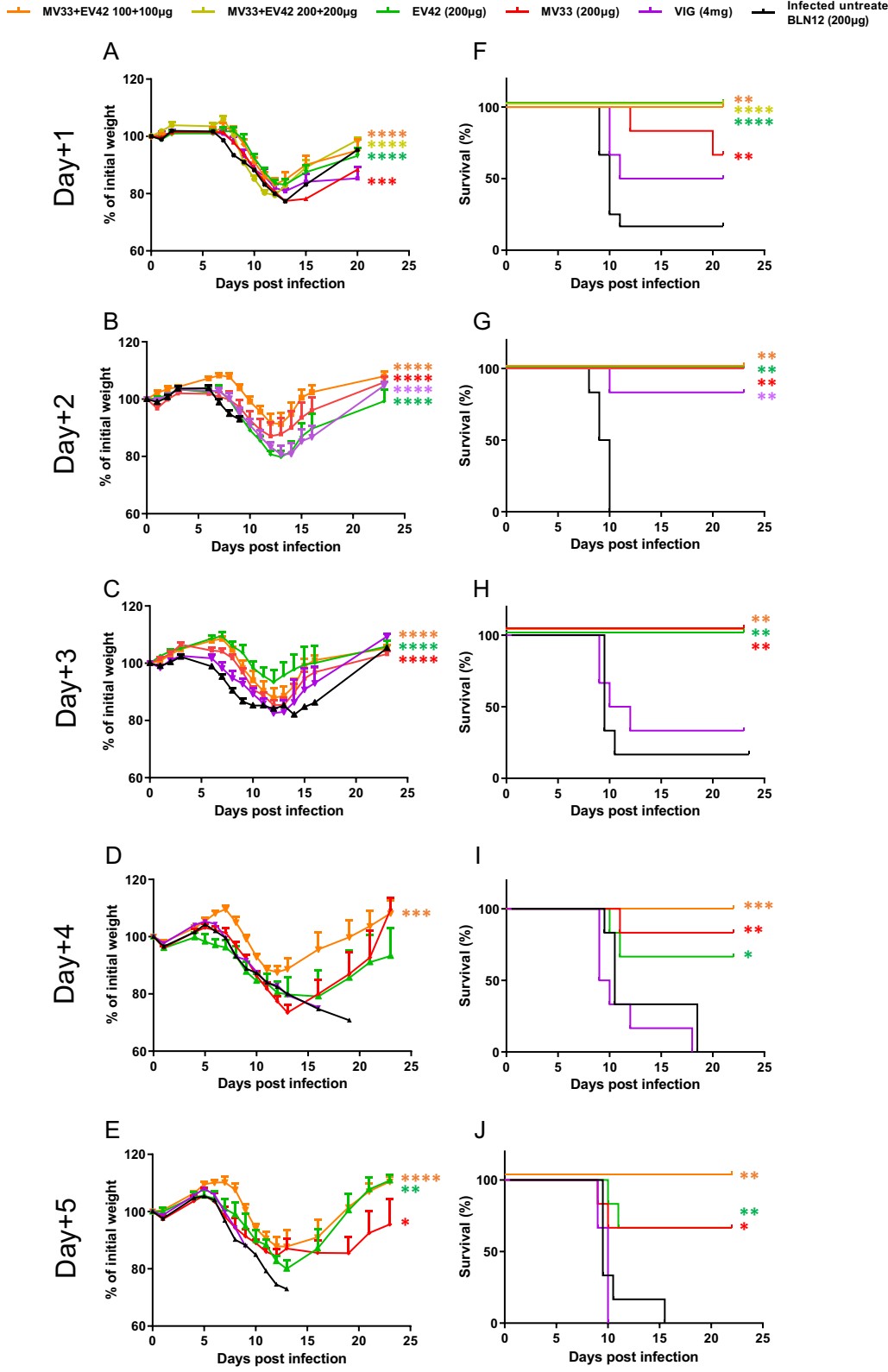

their organs were collected for viral load analysis by plaque assay. As shown in Fig. 4A, C, and E, in the BLN12 treated groups at three dpi, the spleen, liver, and lung already reached very high titers of $1.9 \times 10^8$, $1.6 \times 10^7$ and $8.2 \times 10^7$ PFU/gr tissue, respectively. In the spleen, treatment with MV33 or EV42 alone at three dpi resulted in a nearly one-log and two-log reduction in viral titers ($1.9 \times 10^7$ PFU/gr and $2.4 \times 10^6$ PFU/gr, respectively) (Fig. 4A). However, a significant five-log

reduction ($4 \times 10^2$ PFU/gr) to nearly undetectable levels was observed in the MV33 + EV42 treated group (Fig. 4A). In the liver, the effect was even more pronounced (Fig. 4C). The MV33 treated group resulted in almost undetectable levels ($7.6 \times 10^2$ PFU/gr), while virus was undetectable in either the EV42 or MV33 + EV42 treated groups. Treatment at five dpi was less effective than treatment at three dpi in reducing viral loads at this time point (eight dpi). However, the

**Fig. 1 | Therapeutic efficacy of single dose mAb treatment against lethal ECTV infection.** Body weight changes (**A**–**E**) and survival rates (**F**–**J**) of BALB/c mice infected with ECTV (50 PFU i.n.) and treated with either MV33, EV42, MV33 + EV42, VIGIV, BLN12 (isotype control) or left untreated (infected untreated) at 1–5 days post infection (group legend at the top). *n* = 6 for each experimental group except Day + 1 for groups MV33, EV42, MV33 + EV42 (200 + 200 μg) and infected untreated (*n* = 12) that were combined from two experiments; Statistical significance for (**A**–**E**) was determined by one-way ANOVA of area under the curve with multiple comparisons (color coded; relative to infected untreated group/BLN12).

(**A**, ***$P$ = 0.0001; ****$P$ < 0.0001; **B**, **C**, ****$P$ < 0.0001; **D**, ***$P$ = 0.0002; **E**, *$P$ = 0.04, **$P$ = 0.009, ****$P$ < 0.0001) (**A**–**E**). For (**F**–**J**) Mantel−Cox (Log-ranked) test was used (**F**, MV33 **$P$ = 0.0014; EV42 and MV33 + EV42 (200 + 200μg) ****$P$ < 0.0001; MV33 + EV42 (100 + 100μg) **$P$ = 0.002; **G**, MV33, EV42 and MV33 + EV42 **$P$ = 0.001, VIG **$P$ = 0.004; **H** **$P$ = 0.004; **I**, MV33 **$P$ = 0.005, EV42 *$P$ = 0.03, MV33 + EV42 ***$P$ = 0.0008; **J**, MV33 *$P$ = 0.02, EV42 **$P$ = 0.007, MV33 + EV42 ***$P$ = 0.0004. For (**A**–**E**), measurement data are expressed as mean + standard error (SE). Source data are provided as a Source data file.

combination of MV33 + EV42 resulted in a significant one-log reduction in the spleen (Fig. 4B) and a five-log reduction in the liver (Fig. 4D), reaching undetectable levels. Surprisingly, no significant reduction in viral loads was observed in the lungs of all treated groups (Fig. 4E, F). When comparing these results to the BLI data (Figs. 2 and 3), it is important to clarify that the bioluminescence images are presented at a certain dynamic range and thus do not provide the full and precise data as observed with the viral load analysis.

We further determined whether the combined effect of MV33 + EV42 antibodies is additive or synergistic. To that aim, we calculated the average viral load reduction of each antibody separately or the combination of both, as compared with the isotype control group. We considered a synergistic effect when the fold change of the reduction of both antibodies is greater than the sum of the fold-change reduction of each antibody alone. For example, when mice were treated at three dpi, the viral load in the spleen of MV33 or EV42 treated mice was reduced by 10 or 80-fold, respectively, while the combination treatment resulted in a $4.7 \times 10^3$-fold reduction (Fig. 4A). This synergistic effect was clearly evident in the spleen and liver of mice treated at three or five dpi, respectively (Fig. 4A, D). These results further indicate a synergistic mode of action of both MV33 + EV42 mAbs.

### Treatment with anti D8 and anti A33 antibodies protects against MPXV challenge in mice

Our observation that both anti-D8 and anti-A33 mAbs conferred protection against ECTV infection, along with their demonstrated ability to neutralize several other orthopoxviruses in vitro[13], motivated us to assess, as a proof of concept, their potential protective effects in vivo against MPXV, in light of its current global viral threat. Since commonly used laboratory mice strains like BALB/c are resistant to MPXV, we utilized Cast/EiJ mice, which were previously shown to be sensitive to previous isolates of MPXV exhibiting weight loss and death following MPXV infections[18]. Mice were intranasally challenged with $3.5 \times 10^6$ PFU/mouse of MPXV (clade IIb) and treated at two dpi with MV33 + EV42 (100 μg each). As shown in Fig. 5A, B, in the untreated group, all infected mice displayed morbidity, with a mortality rate of 75%. However, in the treated group, all mice survived, and no signs of morbidity could be observed. We then attempted to further evaluate the therapeutic window in this model. To that aim, mice were either infected or infected and treated at 3–5 dpi. As seen in Fig. 5C, all infected untreated mice start to lose weight at around day five post infection, resulting in a mortality rate of 35%. Treated mice at three dpi were significantly less morbid (Fig. 5C) and were fully protected (Fig. 5D). These positive effects on morbidity were also observed, although to a lesser extent, when treatment was administered at four or five dpi (Fig. 5C). These results further highlight the relevance of our mAbs treatment, as an efficient treatment against MPXV infection as well.

## Discussion

The use of mAbs as a passive treatment approach against viral infection has been widely applied, showing promising results[19]. One major advantage of using mAbs over the use of polyclonal antibodies as therapeutics is their profound and exceptional affinity and high specificity. However, this epitope-specific trait can also become a disadvantage as spontaneous mutations of the targeted virus may result in viral resistance to therapy. This is more pronounced in RNA viruses that are prone to higher frequency of mutations[20], yet, the recent worldwide spread of Mpox disease highlighted this risk in DNA viruses as well[21,22]. In the case of poxviruses, as both intracellular and extracellular forms of the virus, namely MV and EV, are infectious and their surface antigens are distinct, there is a need to develop neutralizing antibodies to key epitopes on the surface of both virus forms[23,24]. The development of drugs, antibodies, or drug-antibody combinations are well investigated in the field of viral infections[25]. Numerous studies have revealed the effectiveness of synergistic antibodies combinations against various viral infections, including Ebola virus[26], Henipaviruses[27], SARS-CoV-2[28] and VACV[14]. In the latter study, it has been shown that the combined action of two antibodies, anti-A33, and anti-L1, targeting the two forms of the VACV virus can enhance its neutralization in vitro[14]. The authors hypothesized that the presence of the anti-A33 antibody and complement can lead to the lysis of the EV membrane, which exposes the MV membrane and enables neutralization of the infections MV form by the anti-L1 antibody. The beneficial effect of the combined antibodies to EV and MV forms can be explained, at least in part, by this mechanism. In our study, we provide evidence that while treatment with either the anti-D8 (MV33) or anti-A33 (EV42) antibody alone results in 50% or 66% survival rates, respectively, administering a half dose of both antibodies leads to full protection at four- or five-days post-infection. Similar results were also reflected by viral loads quantitative analysis of the spleen and liver, where treatment with either MV33 or EV42 antibodies partially reduced the viral load, while a combination of both antibodies resulted in a faster clearance of the virus, leading to nearly undetectable levels in both organs and treatment regimens. These findings suggest a synergistic cooperation between these two mAbs, targeting both MV and EV forms of the virus. The particular mechanism underlying the synergistic cooperation of these antibodies is yet to be elucidated.

Surprisingly, viral loads in the lungs, when tested at day eight post infection, remained high in all treated groups, as opposed to viral load reduction in the spleen and liver. These findings were quite unexpected. Nevertheless, a previously published paper on the therapeutic potential of a recombinant VIG (rVIG) comprising 26 Abs against ECTV challenge, showed a similar observation where a single treatment at four or five dpi resulted in a dramatic viral reduction in the liver but no significant effect between control and treated groups in the lungs[4]. A potential explanation for this phenomenon lies in the kinetic of the virus after intranasal inoculation. Our previous research has demonstrated that ECTV intranasal infection leads to an early, extensive proliferation in the lungs, followed by accumulation of the virus in the spleen and liver[11]. Therefore, at three- or five-days post infection (treatment days), the viral titer in the lungs is significantly higher than the titer in the spleen and the liver posing a greater challenge for neutralization by antibodies, especially if the antibodies are provided systemic rather than locally. In addition, complement proteins are produced by the liver and their concentration in the liver is considerably high[29]. Since MV33 antibody was previously shown to act in a complement dependent manner[13], we speculate that the combination of these antibodies and complement proteins in the liver enables a faster and a more efficient viral clearance as compared to the lungs.

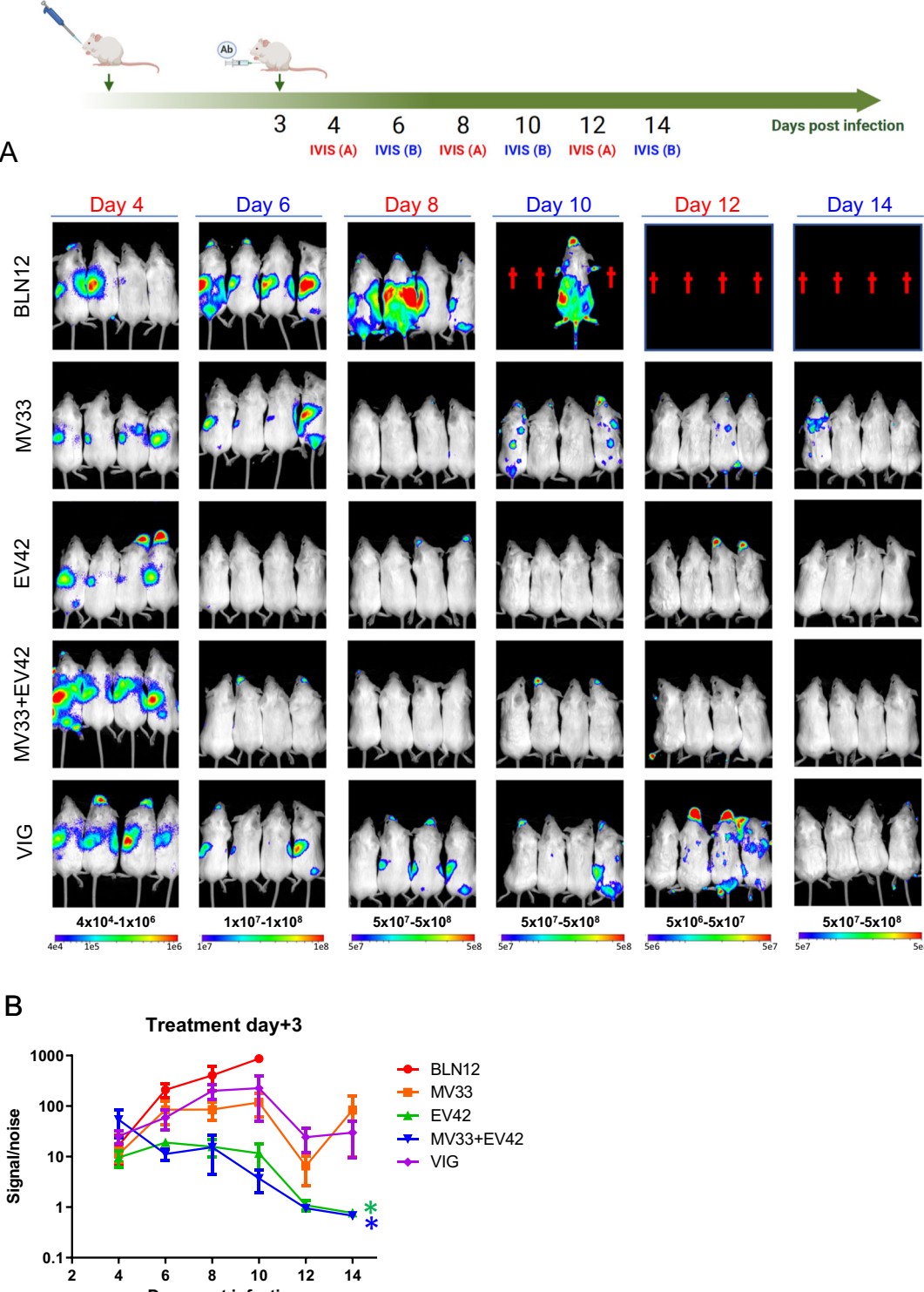

**Fig. 2 | Viral distribution following ECTV infection and treatment at three dpi.**
**A** BALB/c mice were divided into two groups (IVIS (**A**); red and IVIS (**B**); blue) infected with 140 PFU i.n. ECTV-Luc, treated with MV33 (200 µg), EV42 (200 µg), MV33 + EV42 (100 + 100 µg), VIGIV (4 mg) or isotype control (BLN12; 200 µg) on day three p.i. (*n* = 8) and imaged on days 4, 8 and 12 (IVIS (**A**); *n* = 4 for each treated group; red) or on days 6, 10 and 14 (IVIS (**B**); *n* = 4 for each treated group; blue). Bioluminescent signal intensity range is shown below each column (color scale;

photon/s/cm2/sr). Dagger represents dead mice. The Scheme was created with BioRender.com (full license). **B** Bioluminescent signal/noise intensity as total photon flux (photon/s/cm²/sr), was calculated by region of interest (ROI) analysis on the chest and abdomen cavity. Same ROI was used for all mice examined. *$P = 0.04$ One-way ANOVA with multiple comparisons vs. BLN12 (color coded). Measurement data are expressed as mean ± standard error (SE). Source data are provided as a Source data file.

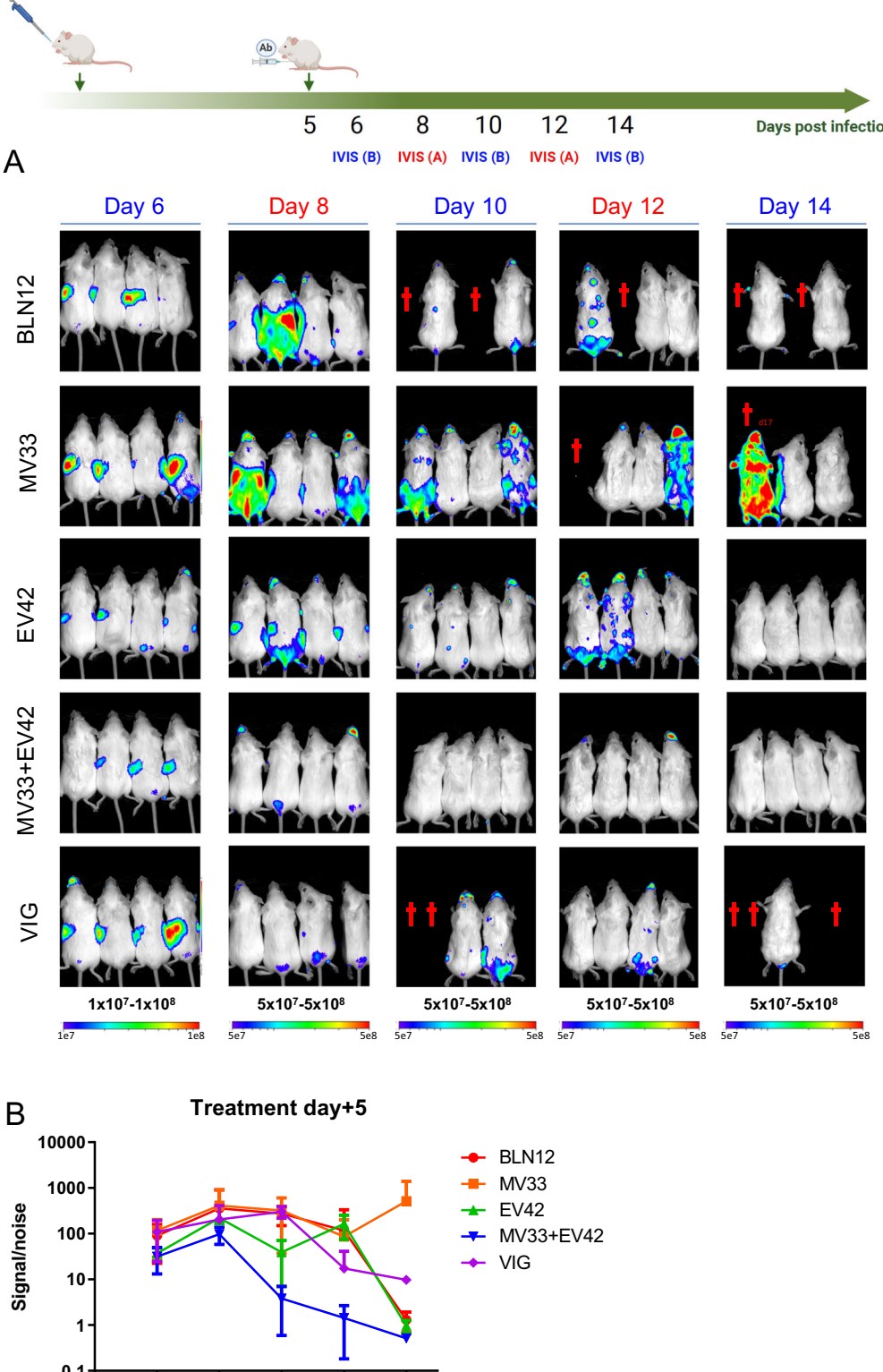

**Fig. 3 | Viral distribution following ECTV infection and treatment at five dpi.**
**A** BALB/c mice were divided into two groups (IVIS (A); red and IVIS (B); blue) infected with 140 PFU i.n. ECTV-Luc, treated with MV33 (200 µg), EV42 (200 µg), MV33 + EV42 (100 + 100 µg), VIGIV (VIG; 4 mg) or isotype control (BLN12; 200 µg) on day five p.i. (n = 8) and imaged on days 6, 10 and 14 (IVIS (**B**) or on days 8 and 12 (IVIS (**A**); n = 4 for each treated group; red); n = 4 for each treated group; blue). Bioluminescent signal intensity range is shown below each column (color scale;

photon/s/cm2/sr). Dagger represents dead mice. The Scheme was created with BioRender.com (full license). **B** Bioluminescent signal/noise intensity as total photon flux (photon/s/cm²/sr), was calculated by region of interest (ROI) analysis on the chest and abdomen cavity. Same ROI was used for all mice examined. Measurement data are expressed as mean ± standard error (SE). Source data are provided as a Source Data file.

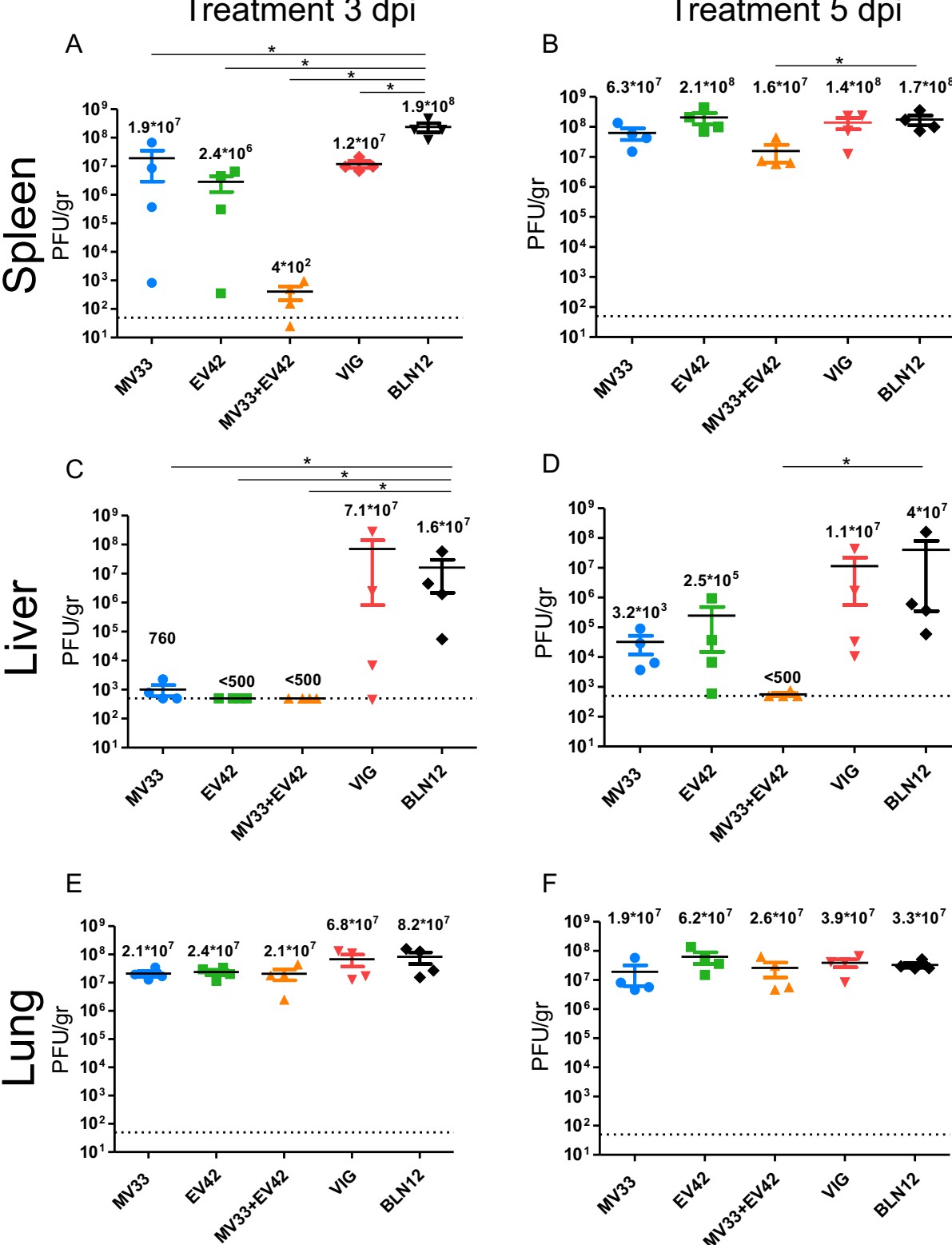

**Fig. 4 | Viral load analysis following ECTV challenge and mAbs treatment.** Viral titers were determined by plaque assay from the indicated organs (spleen, liver, and lung) of BALB/c mice, eight dpi with 50 PFU i.n. ECTV. Viral loads are presented for mice treated at three dpi. (**A**, **C** and **E**) and at five dpi. (**B**, **D** and **F**). Horizontal lines represent the mean of each group (indicated above each line). Dashed line represents limit of detection (LOD). Samples below the LOD assigned a value of half of the LOD. Asterisk denote for significant reduction in viral load compared to the BLN12 (isotype control treated group) using Mann-Whitney non-parametric two-tailed unpaired *T* test (*n* = 4, *P* = 0.03). The error bars represent the SEs. Source data are provided as a Source data file.

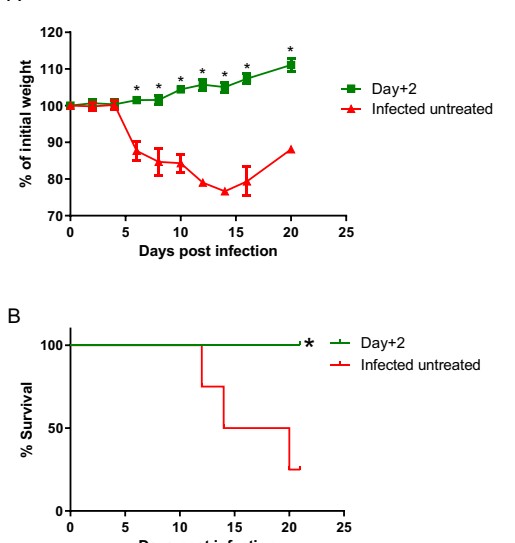

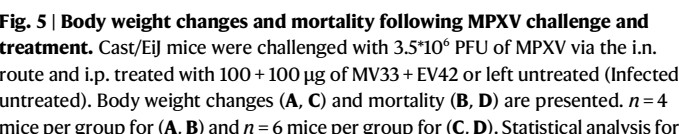

**Fig. 5 | Body weight changes and mortality following MPXV challenge and treatment.** Cast/EiJ mice were challenged with 3.5*10⁶ PFU of MPXV via the i.n. route and i.p. treated with 100 + 100 µg of MV33 + EV42 or left untreated (Infected untreated). Body weight changes (**A**, **C**) and mortality (**B**, **D**) are presented. *n* = 4 mice per group for (**A**, **B**) and *n* = 6 mice per group for (**C**, **D**). Statistical analysis for (**A**, **C**) was performed using two-tailed unpaired *t* test per row, with correction for multiple comparisons using the Holm–Sidak method, *P < 0.003. For (**B**, **D**), Log-rank (Mantel-Cox) Test vs. infected untreated was performed, *P = 0.04. For (**A**, **C**), measurement data are expressed as mean ± standard error (SE). Source data are provided as a Source data file.

Currently, the only two approved antivirals against orthopoxviruses are brincidofovir (BCV, previously CMX001) and tecovirimat (previously ST-246)[30]. However, achieving full protection with these drugs in the ECTV mouse model of smallpox requires multiple administrations. In the present study, we demonstrate that a single administration of a combination of only two monoclonal antibodies (mAbs) provide complete protection against orthopoxviruses, even when administered as late as five days post infection.

The two in vivo mice models for ECTV and MPXV that we have applied in this study, displayed different morbidity and mortality responses following mAbs treatment. ECTV infected and treated mice were morbid with significant weight loss, but with an impressive recovery and survival rates from infection, even upon late treatment (five dpi). Conversely, MPXV infected and treated mice at two dpi gained weight and only slightly lose weight when treatment was administered at three dpi. It should be noted here that despite the underlying similarities between orthopoxviruses allowing for the use of a single vaccine or an antibody treatment against different members of the genus, the viruses differ in their virulence and hosts. As such, ECTV and MPXV infections of mice might generate two different diseases with different manifestations, rates, and affected tissues. Thus, the mAbs therapeutic outcomes may differ with respect to lethal dose, disease kinetics, viral loads at different organs and therapeutic windows. For example, as demonstrated above, the morbidity of MPXV-infected mice starts at an earlier time point (around five dpi) compared to ECTV (eight dpi) ([4] and Figs. 5 and 1, respectively). In any case, these results highlight the potential of the combination of these two antibodies as an efficient treatment against orthopoxviruses including MPXV.

It has been previously shown that both VACV vaccination and MPXV natural infection result in the production of neutralizing antibodies directed against the following proteins: D8, A33, A27, H3, L1, and B5. These antibodies have been shown to confer protection against systemic infection[3,5,31]. These findings highlight the relevance and potential of our anti D8 and anti A33 antibodies as a potential therapy against orthopoxviruses.

It should be noted that these mAbs are expected to exhibit a relatively long half-life in the human body, due to their IgG1 subtype (~21 days[32]), and hence should also be suitable for prophylactic treatment. Also, they can be further manipulated, by the addition of distinct mutation to their Fc region, to extend their half-life up to 90 days[33–35]. In summary, this work describes an in-depth in vivo strategy by which a combination therapy of two human-like monoclonal antibodies act synergistically to provide superior protection against ECTV infection in mice. This combination of two mAbs has the potential to be used therapeutically against orthopoxvirus infections, and may provide an alternative to VIGIV.

## Methods
### Cells and viruses
BS-C-1 (ATCC CCL-26) and HeLa (ATCC CCL-2) cells were used and were grown in growth medium [Dulbecco's Modified Eagle's Medium (DMEM) containing 10% fetal bovine serum (FBS), MEM nonessential amino acids (NEAA), 2 mM L-glutamine, 100 Units/ml penicillin, 0.1 mg/ml streptomycin, 12.5 Units/ml nystatin (P/S/N), all from Biological Industries, Israel]. Cells were cultured at 37 °C, 5% CO2 with 95% humidity. ECTV expressing firefly luciferase (ECTV-Luc)[36] was kindly provided by Prof. Luis Sigal, Thomas Jefferson University hospital, Philadelphia, USA. Briefly, ECTV Moscow (ATCC VR-1374) and ECTV-Luc were propagated in HeLa cells and titrated on BS-C-1 cells[37]. MPXV 2018 (accession no. MN648051)[38], was grown in HeLa cells, and titer was determined in BS-C-1 cells[13].

### Animal challenge experiments
All animal experiments in this study were approved by the Institutional Animal Care and Use Committee of the Israel Institute For Biological Research (IIBR). Experimental procedures were performed under Protocol Numbers M-53-21, M-25-22, M-34-22, and M-67-02-19 (MPXV). Handling and working with ECTV and MPXV samples were conducted in a BSL3 facility in accordance with the biosafety guidelines of the Israel Institute for Biological Research (IIBR). All viral challenges were performed by intranasal instillation (i.n.) and all antibody treatments were given by intraperitoneal route.

ECTV and ECTV-Luc experiments - Female BALB/c mice (six–eight weeks old, 15–18 gr.) were purchased from Charles River Laboratories, Margate, UK. Mice were acclimatized for a week prior to the experiment. Mice were anesthetized and challenged i.n. with 50 PFU ECTV (1 PFU = 1 LD50) or 140 PFU ECTV-Luc (7 PFU = 1 LD50).

For LD50 evaluation experiment of ECTV and ECTV-Luc, mice were anesthetized and challenged i.n. with 0.1, 1, 10 or 100 PFU.

Antibodies treatment - Mice were treated with a single dose of MV33 (200 μg), EV42 (200 μg), combination of both (100 + 100 μg or 200 + 200 μg), commercial VIGIV (50 mg/ml) by Omrix Biopharmaceuticals Ltd. (Omr IgG-am 5% VIG) (4 mg) or mAb isotype control (BLN12, anti-SARS-CoV-2)[39] at day 1 to 5 post infection (n = 6 for each experimental group).

MPXV experiment - Female Cast/EiJ mice (8–10 weeks, 10–15 gr.) were purchased from Jackson Laboratories (Bar Harbor, ME, USA) and acclimatized under supervision for a week prior to the experiment. Mice were challenged (i.n.) with $3.5 \times 10^6$ PFU MPXV (1-2 LD50) (2018; clade IIb), treated with a single dose of MV33 + EV42 (100 + 100 μg) at 2–5 dpi or left untreated (n = 4 for 2 dpi or n = 6 for 3–5 dpi).

## Bioluminescence imaging

Live imaging was performed with an IVIS Lumina II system (Caliper LifeSciences, MA). D-Luciferin substrate (Caliper LifeSciences, MA) was injected intraperitoneally (i.p.) (150 μg/g body weight) 7 min prior to imaging. Mice were imaged under anesthesia with Ketamine 75 mg/kg, Xylazine 7.5 mg/kg in PBS. Group A- animals were imaged on days 4, 8, 12 post infection (for mice treated at three dpi) or 8 and 12 post infection (for mice treated at five dpi). Group B – animals were imaged on days 6, 10 and 14 post infection (for both three and five dpi-treated mice). Images were collected for 1 or 40 s with binning factor of 4. Bioluminescent signal/noise intensity as total photon flux (photon/s/cm²/sr) was calculated by region of interest (ROI) analysis on the chest and abdomen cavity (for all mice examined). Same ROI was used in all examined mice for calculation of signal intensity (from the chest through the hind limbs). Acquisition and analysis were performed with Living Image Software, Version 4.2 (Calliper LifeSciences, Hopkinton, MA)[17].

## Viral load in mouse organs

Viral loads of ECTV in spleen, liver and lung were determined in mice 8 days post infection. Animals were anesthetized and sacrificed. Organs were harvested, transferred immediately to liquid nitrogen and stored at −80 °C until further processing. Organs were homogenized (ULTRA-TURAX® IKA R104) for 30 sec in ice cold PBS (1 ml for spleen and lungs and 4 ml for liver), centrifuged (1300 rpm, 10 min, 4 °C) and supernatants were collected for virus titration. Titration of ECTV was performed on 100% confluent monolayers of BSC-1 cells in 12 well tissue culture grade plates (Nunc). ECTV viral load was determined using the plaque-forming unit (PFU) assay. Serial dilutions of extracted organs were prepared in infection medium (MEM containing 2% fetal calf serum (FCS), L-glutamine, non-essential amino-acids solution and penicillin-streptomycin solution (Biological Industries, Israel)) and used to infect BS-C-1 monolayers in duplicates (200 μL/well). Plates were incubated for 1 h at 37 °C to allow viral adsorption. Then, 2 mL/well of overlay (5% W/V methyl cellulose (Sigma)) was added to each well, and plates were incubated at 37 °C, 5% CO2, for 5 days. The medium was then aspirated, and the cells were fixed and stained for 5 min at room temp. with 1 mL/well of crystal violet solution (Biological Industries, Beit-Haemek, Israel). The number of plaques in each well was determined, and ECTV viral titer was calculated.

## Statistical analyses

Statistical analyses were performed using GraphPad Prism 6. Exact p values are provided for each analysis. The analysis for morbidity experiments performed using the area under the curve (AUC) for each animal. Mean AUC's of the various groups were compared using one-way ANOVA with multiple comparisons or two-tailed unpaired t test. For survival experiments, Kaplan-Meier survival plots were compared by the Cox-Mantel test. For viral load analysis, Mann-Whitney non-parametric two-tailed unpaired t test was used. For bioluminescent intensity analysis, one-way ANOVA was used. A value of P < 0.05 was accepted as statistically significant.

## Reporting summary

Further information on research design is available in the Nature Portfolio Reporting Summary linked to this article.

## Data availability

All data that support the findings are available in the main text and supplementary material. The antibodies are available (by contacting T.I. from the Israel Institute For Biological Research; tomeri@iibr.gov.il) for research purposes only under an MTA, which allows the use of the antibodies for non-commercial purposes but not to its disclosure to third parties. Source data are provided in this paper.

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

## Acknowledgements
We acknowledge the IIBR administrative personnel for their commitment to the project. We thank Prof. Luis Sigal for providing the ECTV-Luc. We thank Dr. Amir Rosner, Michael Yampolski, Tima Glakin, and Yossi Schlomovitch for animal husbandry.

## Author contributions
H.T., T.N.P., S.M., L.C.M., M.B.G., B.P., N.E., N.P., and T.I. conceived and designed the experiments. T.N.P., R.A., H.A., S.W., R.R., E.E., O.M., and E.M. contributed in technical support. H.T., Y.Y.R., H.A., N.P., and T.I. analyzed the data. H.T., T.N.P., Y.Y.R., N.P., and T.I. wrote the manuscript. All authors commented on and approved the manuscript.

## Competing interests
Patent application (PCT/IL2024/050198) for the described antibodies was filed by the Israel Institute for Biological Research. The patent covers the sequence and use of these antibodies. There is no restriction on data publication. H.T., T.N.P., S.M., N.P., E.M., R.E, R.R, O.M and T.I. are inventors of this patent application. The remaining authors declare no competing interests.
