## [Peer Review File · Nature Communications]

REVIEWER COMMENTS

Reviewer #1 (Remarks to the Author):

Review of “Synergistic effect of two human-like monoclonal antibodies confers protection against orthopoxvirus infection” by Tamir et al. This manuscript comprises of the assessment of two mAbs targeting different Orthopox antigens and their use within an animal model using ECTV.

There are no line numbers, so this has made the review rather difficult to refer to specific lines.

Minor comments:

Introduction-

When referring to Orthopox viruses, there is a tendency of the authors to miss out ‘virus’ when shortening the viral name, e.g. “Ectromelia (ECTV)” but should instead read as “Ectromelia virus (ECTV)”, the same is for VARV, VACV and MPXV

“ECTV, similarly to VARV, encodes multiple host-specific immunoregulatory genes and causes severe lethal disease associated with multi organ high viral loads and various manifestations” – some references should be provided here

Discussion-

The authors describe utility of a number of different drug combinations for different viruses, however are focussed on drugs such as Favipiravir and Oseltamivir, which is irrelevant considering the article is focussed on mAbs.

The authors should expand on the utility of these mAbs targeting A33 and D8, do we see normal vaccination inducing antibodies to these antigens? And does either VACV or MPXV infection induce similar antibodies?

What is the longevity of the antibodies when given? Considering that these mAbs offer protection up until 5 dpi, could they be offered as a prophylaxis? Further discussion on this is needed, especially in the context of long lasting RSV mAbs

Major comments:

Results-

The authors state “When the mAb treatment using both MV33+EV42, was provided on day 6 pi, no protection in terms of morbidity or survival was observed (data not shown)” – this is data that should be included within this manuscript, as it is important and of use to other researchers

At the beginning of page 7, the authors state “Mice were infected with a lethal dose of 140 PFU ECTV-Luc...”, why is there an almost 3x dose used compared to the results for Fig 1? The authors previously state a lethal dose of 50 PFU? What is the reason for 3x higher dosage was used for one experiment compared to others, this raises some concern with the experiments listed with differing doses and comparisons of results. There is no clarification in the discussion about this difference as well which needs to be addressed.

Figure 4 – additional statistical annotations should be used here, with clarity on exact p-values needed

“Mice were intranasally challenged with 3.5×10^6 PFU/mouse of MPXV (clade IIb) and treated at 2 dpi with MV33+EV42 (100 µg each).” – why did the authors not also conduct mAb treatment at 3 or 5 dpi? As this data would provide comparability with ECTV work observed and identify if administration is OPXV-independent and more specifically about timing?

Methods – clarity on how the mice were challenged is needed – some experiments say intranasal, others say intraperitoneal

Reviewer #2 (Remarks to the Author):

This is a well written manuscript demonstrating the benefits of a 2 mAb combination for orthopoxvirus infection using two mouse models. The extension of the therapeutic window demonstrated in the ECTV model compared to previous work cited by the authors is noteworthy and significant. The results are presented in a clear manner and the conclusions appropriately justified.

I would disagree with the author's blanket assertion in the Discussion: "...when developing therapeutic preparations, it is crucial to include at least two different mAbs targeting different epitopes, to overcome the potential of viral resistance." There are many factors in play in choosing composition of a anti-viral therapeutic including the propensity of a virus to be able to escape from a specific mAb. As just one example, I would prefer a single mAb therapeutic that has been shown to be difficult for a virus to escape from then a three mAb cocktail where escape can occur readily to the individual mAbs.

The findings that viral load in the lungs was not affected by mAb treatment were very unexpected. The authors cite examples of this phenomenon with small molecules but not for neutralizing mAbs. Some additional speculation on the root cause of this unusual observation would be welcome.

Reviewer #1 (Remarks to the Author):

Review of “Synergistic effect of two human-like monoclonal antibodies confers protection against orthopoxvirus infection” by Tamir et al. This manuscript comprises of the assessment of two mAbs targeting different Orthopox antigens and their use within an animal model using ECTV.

There are no line numbers, so this has made the review rather difficult to refer to specific lines.

Line numbering was added to the manuscript.

Minor comments:

Introduction-

When referring to Orthopox viruses, there is a tendency of the authors to miss out ‘virus’ when shortening the viral name, e.g. “Ectromelia (ECTV)” but should instead read as “Ectromelia virus (ECTV)”, the same is for VARV, VACV and MPXV

The text was corrected in the revised manuscript (lines 39 and 61).

“ECTV, similarly to VARV, encodes multiple host-specific immunoregulatory genes and causes severe lethal disease associated with multi organ high viral loads and various manifestations” – some references should be provided here

The following references were added:

9. Garver, J., et al., Ectromelia Virus Disease Characterization in the BALB/c Mouse: A Surrogate Model for Assessment of Smallpox Medical Countermeasures. *Viruses*, 2016. 8(7).
10. Mavian, C., et al., Comparative Pathogenesis, Genomics and Phylogeography of Mousepox. *Viruses*, 2021. 13(6).
11. Paran, N., et al., Postexposure immunization with modified vaccinia virus Ankara or conventional Lister vaccine provides solid protection in a murine model of human smallpox. *J Infect Dis*, 2009. 199(1): p. 39-48.
12. Sakala, I.G., et al., Evidence for Persistence of Ectromelia Virus in Inbred Mice, Recrudescence Following Immunosuppression and Transmission to Naive Mice. *PLoS Pathog*, 2015. 11(12): p. e1005342.

Discussion-

The authors describe utility of a number of different drug combinations for different viruses, however are focussed on drugs such as Favipiravir and Oseltamivir, which is irrelevant considering the article is focussed on mAbs.

As suggested, the irrelevant paragraph was omitted from the text (lines 295-300).

The authors should expand on the utility of these mAbs targeting A33 and D8, do we see normal vaccination inducing antibodies to these antigens? And does either VACV or MPXV infection induce similar antibodies?

“do we see normal vaccination inducing antibodies to these antigens?”

Antibody response characterization to VACV in immunized volunteers has been previously published (Lantto et al. 2011). In their study, they have isolated and characterized the following specific VACV antigens: A10L, A14L, A27L, A33R, A56R, B5R, D8L, H3L, H5R, I1L, VCP, and the ATI protein. Moreover, in our recent publication in Microbiology spectrum (Noy-Porat et al. 2023) we have analyzed the antibody specificity profiles of the two VACV immunized NHP sera using an array of the 224 VACV proteins. Both A33 and D8 were among the top 10 proteins with the highest response to both sera.

“And does either VACV or MPXV infection induce similar antibodies?” Several studies have shown that both VACV and MPXV infections resulted in the production of neutralizing antibodies against similar proteins including D8, A33, A27, L1, H3, L1 and B5. These antibodies have been shown to confer protection against systemic infection (Sakhatsky P et al. 2006; Gilchuk et al, 2016).

The following section was added to the text (lines 368-373) – “It has been previously shown that both VACV vaccination and MPXV natural infection result in the production of neutralizing antibodies directed against the following proteins: D8, A33, A27, L1, H3, L1 and B5. These antibodies have been shown to confer protection against systemic infection [3, 5, 33]. These findings highlight the relevance and potential of our anti D8 and anti A33 antibodies as a potential therapy against orthopoxviruses. “

What is the longevity of the antibodies when given? Considering that these mAbs offer protection up until 5 dpi, could they be offered as a prophylaxis? Further discussion on this is needed, especially in the context of long lasting RSV mAbs

Indeed, evaluating the prophylaxis effect of these Abs could be a good option. Both antibodies are of the IgG1 subtype, which exhibits a half-life of approximately 21 days in the human body (Chames et al. 2009; Ref #35). These mAbs therefore can be suitable for prophylactic treatment as well.

We have added a section in the discussion regarding this issue (lines 374-378) – “It should be noted that these mAbs are expected to exhibit a relatively long half-life in the human body, due to their IgG1 subtype (approximately 21 days; [35]), and hence should also be suitable for prophylactic treatment. Also, they can be further manipulated, by the addition of distinct mutation to their Fc region, to extend their half-life up to 90 days [36-38].”

Major comments:

Results-

The authors state “When the mAb treatment using both MV33+EV42, was provided on day 6 pi, no protection in terms of morbidity or survival was observed (data not shown)” – this is data that should be included within this manuscript, as it is important and of use to other researchers

As suggested, data on the morbidity and mortality of mice treated with both abs at 6 dpi was added to the manuscript (Supplementary Figure 1) as referred in the text (line 122).

Supplementary Figure 1. Therapeutic efficacy of single dose mAb treatment against lethal ECTV infection at 6 days post infection. Body weight changes (A) and survival rates (B) of BALB/c mice infected with ECTV (50 PFU i.n.) and treated with MV33+EV42 (100+100µg) or left untreated (infected untreated). n = 6 / group.

At the beginning of page 7, the authors state “Mice were infected with a lethal dose of 140 PFU ECTV-Luc...”, why is there an almost 3x dose used compared to the results for Fig 1? The authors previously state a lethal dose of 50 PFU? What is the reason for 3x higher dosage was used for one experiment compared to others, this raises some concern with the experiments listed with differing doses and comparisons of results. There is no clarification in the discussion about this difference as well which needs to be addressed.

A clarification regarding this concern has been added to the revised manuscript (lines 140-143; 410-411).

Lines 140-143: “Compared to the parental ECTV used above, ECTV-Luc is slightly less virulent [17] as seen in supplementary Figure 2. Hence, to achieve 20 LD₅₀, a dose of 140 PFU was used, instead of the 50 PFU used with the parental virus.”

Lines 410-411: “For LD₅₀ evaluation experiment of ECTV and ECTV-Luc, mice were anesthetized and challenged i.n. with 0.1, 1, 10 or 100 PFU.”

A supplementary figure was added (Supplementary Figure 2).

LD₅₀ calculated ratio: 4.02 pfu (ECTV-Luc) / 0.58 pfu (ECTV) = 7

Supplementary Figure 2. **Lethal dose evaluation of ECTV-Luc and ECTV.** Survival rates of BALB/c mice infected with either ECTV-Luc (A) or ECTV (B) at the indicated doses. The LD₅₀ values for each strain and the ratio between these values was calculated.

Figure 4 – additional statistical annotations should be used here, with clarity on exact p-values needed

Additional statistical annotations were added to figure legend (lines 240-242): “Asterisk denote for significant reduction in viral load compared to the BLN12 (isotype control treated group) using Mann-Whitney non-parametric unpaired T-test (n=4, *p=0.03). The lines represent the means. The error bars represent the SEs. ”

Clarity on the p-values needed was added to the Methods section: “A value of p<0.05 was accepted as statistically significant.” (lines 459-460).

“Mice were intranasally challenged with 3.5×10^6 PFU/mouse of MPXV (clade IIb) and treated at 2 dpi with MV33+EV42 (100 µg each).” – why did the authors not also conduct mAb treatment at 3 or 5 dpi? As this data would provide comparability with ECTV work observed and identify if administration is OPXV-independent and more specifically about timing?

As suggested by the reviewer, we have further evaluated the therapeutic window of the two mAbs against MPXV at 3-5 dpi. This additional experimental data was added to the text and to the revised Figure 5 (lines 257-264; 347-367).

Figure 5. Body weight changes and mortality following MPXV challenge and treatment. Cast/EiJ mice were challenged with 3.5×10^6 PFU of MPXV via the i.n. route and i.p. treated with 100+100µg of

MV33+EV42 or left untreated (Infected untreated). Body weight changes (A, C) and mortality (B, D) are presented. n=4 mice per group for A, B and n=6 mice per group for C, D. Statistical analysis for A, C was performed using one unpaired *t*-test per row, with correction for multiple comparisons using the Holm-Sidak method, $p < 0.002$ or Log-rank (Mantel-Cox) Test vs. infected untreated (* $p < 0.05$.) for B, D. For (A, C), measurement data are expressed as mean \pm standard error (SE).

Methods – clarity on how the mice were challenged is needed – some experiments say intranasal, others say intraperitoneal

All viral challenges were performed by intranasal inoculation while all antibody treatments were given by intraperitoneal route. Clarification on this issue was added to the method section in the revised manuscript (lines 401-403).

Reviewer #2 (Remarks to the Author):

This is a well written manuscript demonstrating the benefits of a 2 mAb combination for orthopoxvirus infection using two mouse models. The extension of the therapeutic window demonstrated in the ECTV model compared to previous work cited by the authors is noteworthy and significant. The results are presented in a clear manner and the conclusions appropriately justified.

We thank the reviewer for his positive feedback.

I would disagree with the author's blanket assertion in the Discussion: "...when developing therapeutic preparations, it is crucial to include at least two different mAbs targeting different epitopes, to overcome the potential of viral resistance." There are many factors in play in choosing composition of an anti-viral therapeutic including the propensity of a virus to be able to escape from a specific mAb. As just one example, I would prefer a single mAb therapeutic that has been shown to be difficult for a virus to escape from then a three mAb cocktail where escape can occur readily to the individual mAbs.

We agree with this comment regarding viruses in general. However, due to the complex nature of poxviruses, comprising of two distinct forms, each exhibiting different surface antigens, it is considered important to include antibodies targeting both forms.

As suggested, the general statement was omitted from the text (lines 286-288).

The findings that viral load in the lungs was not affected by mAb treatment were very unexpected. The authors cite examples of this phenomenon with small molecules but not for neutralizing mAbs. Some additional speculation on the root cause of this unusual observation would be welcome.

The following suggested hypotheses were added to the discussion section (lines 319-339). The references regarding this phenomenon with small molecules were omitted (lines 336-339).

“These findings were quite unexpected. Nevertheless, a previously published paper on the therapeutic potential of a recombinant VIG (rVIG) comprising 26 Abs against ECTV challenge, showed a similar observation where a single treatment at 4 or 5 dpi resulted in a dramatic viral reduction in the liver but no significant effect between control and treated groups in the lungs [4]. A potential explanation for this phenomenon lies in the kinetic of the virus after intranasal inoculation. Our previous research has demonstrated that ECTV intranasal infection leads to an early, extensive proliferation in the lungs, followed by accumulation of the virus in the spleen and liver [11]. Therefore, at 3- or 5-days post infection (treatment days), the viral titer in the lungs is significantly higher than the titer in the spleen and the liver posing a greater challenge for neutralization by antibodies, especially if the antibodies are provided systemic rather than locally. In addition, complement proteins are produced by the liver and their concentration in the liver is considerably high [30]. Since MV33 antibody was previously shown to act in a complement dependent manner [13], we speculate that the combination of these antibodies and complement proteins in the liver enables a faster and a more efficient viral clearance as compared to the lungs.”

REVIEWERS' COMMENTS

Reviewer #2 (Remarks to the Author):

The issues raised by reviewers were well addressed.